# Occupational Safety and Health Staging Framework for Decent Work

**DOI:** 10.3390/ijerph191710842

**Published:** 2022-08-31

**Authors:** Paul A. Schulte, Ivo Iavicoli, Luca Fontana, Stavroula Leka, Maureen F. Dollard, Acran Salmen-Navarro, Fernanda J. Salles, Kelly P. K. Olympio, Roberto Lucchini, Marilyn Fingerhut, Francesco S. Violante, Mahinda Seneviratne, Jodi Oakman, Olivier Lo, Camila H. Alfredo, Marcia Bandini, João S. Silva-Junior, Maria C. Martinez, Teresa Cotrim, Folashade Omokhodion, Frida M. Fischer

**Affiliations:** 1Advanced Technologies and Laboratories International, Inc., Gaithersburg, MD 20878, USA; 2Section of Occupational Medicine, Department of Public Health, University of Naples Federico II, 80131 Naples, Italy; 3Business School, University College Cork, T12 K8AF Cork, Ireland; 4PSC Global Observatory, University of South Australia, Adelaide, SA 5000, Australia; 5Department of Population Health, NYU Grossman School of Medicine, New York, NY 10016, USA; 6Department of Environmental Health, School of Public Health, University of São Paulo, São Paulo 01246-904, Brazil; 7Environmental Health Sciences, School of Public Health, Florida International University, Miami, FL 33199, USA; 8Occupational Medicine, Department of Medical and Surgical Specialties, Radiological Sciences and Public Health, University of Brescia, 25121 Brescia, Italy; 9Department of Medical and Surgical Sciences, University of Bologna, 40138 Bologna, Italy; 10Hygiene and Toxicology Team, Safe Work NSW, Parramatta, NSW 2153, Australia; 11Center for Ergonomics and Human Factors, LaTrobe University, Melbourne, VIC 3086, Australia; 12Medical Services Division, International SOS, Singapore 486018, Singapore; 13Department of Public Health, School of Medicine, University of Campinas, Campinas 13083-970, Brazil; 14Medical School, University of São Paulo, São Paulo 04263-200, Brazil; 15WAF Informatics & Health Ltd., São Paulo 04109-100, Brazil; 16Ergonomics Laboratory, CIAUD, Faculdade de Motricidade Humana, University of Lisbon, 1499-002 Lisbon, Portugal; 17Division of Occupational Health, College of Medicine, University of Ibadan, Ibadan 200132, Nigeria

**Keywords:** psychosocial hazards, climate change, migrant workers, technology

## Abstract

The 2030 United Nations Goal 8 for sustainable development focuses on decent work. There is utility in identifying the occupational safety and health aspects of Goal 8, as they pertain to the four pillars of decent work: job creation, social protection, rights of workers, and social dialogue. A workgroup of the International Commission on Occupational Health and collaborators addressed the issue of decent work and occupational safety and health (OSH) with the objective of elaborating a framework for guidance for practitioners, researchers, employers, workers, and authorities. This article presents that framework, which is based on an examination of the literature and the perspectives of the workgroup. The framework encompasses the intersection of the pillars of decent (employment creation, social protection, rights of workers, and social dialogue) work with new and emerging hazards and risks related to various selected determinants: new technologies and new forms of work; demographics (aging and gender); globalization; informal work; migration; pandemics; and OSH policies and climate change. The OSH field will need an expanded focus to address the future of decent work. This focus should incorporate the needs of workers and workforces in terms of their well-being. The framework identifies a starting point for the OSH community to begin to promote decent work.

## 1. Introduction

The inherent characteristics of work, the workforce, and the workplace are continuously changing, but at a faster rate than in the past [1]. These changes are due to a multiplicity of factors, such as demographics, social and economic changes, technology, and, more recently, pandemics. What should the occupational safety and health (OSH) community consider to promote decent work for workers in these changing conditions? Decent work, a United Nations (UN) goal, is employment that respects the “…*fundamental rights of workers in terms of conditions of work safety, health, and remuneration*…”, as well as respects the physical and mental integrity of the worker in the exercise of his or her employment [2]. “*All countries stand to benefit from having a healthy, well-educated workforce with the knowledge and skills needed for productive, fulfilling work and full participation in society* (para. 27, [2])”.

The Decent Work Agenda was launched by the International Labour Organization (ILO) in 1999 and later became part of the UN agenda for sustainable development [2,3] as Goal 8, recognizing that decent work supports inclusive and sustainable economic growth and full, productive employment. The OSH community now needs a comprehensive approach for addressing the profound changes in the world of work and meeting UN Goal 8. Here, we take initial steps toward meeting that need by providing a framework to identify the OSH elements within decent work. To that end, a diverse group of experienced authors examined the relationship between critical determinants and pillars of decent work and OSH. This article is a commentary based on the selected literature identified by the authors as pertinent to the various topics.

The framework in this paper is built on the recognition that the field of OSH must evolve to meet the needs of workers and the workforce in the future [4,5,6,7,8,9,10,11,12,13,14]. The framework encompasses factors that influence worker health and well-being that “…*go beyond traditional OSH concerns (exposures to chemical, physical, or biological agents as well as work-related stress risks) and account for emerging demographic profiles (e.g., more women, immigrants, and older workers, and more chronic disease and mental health conditions), varying employment arrangements*…[4]”, increasing work demands, increasing psychosocial hazards, political and economic influences, and changing work environments (built and natural) [6,15,16,17,18,19,20,21,22,23,24].

The rationale for the framework has precedents, such as the “…World Health Organization (WHO) Healthy Workplaces global model for action, various European efforts for well-being, and the U.S. National Institute for Occupational Safety and Health (NIOSH) Total Worker Health^®^ perspective…[4],” which all aim to protect workers, prevent injury and disease, and promote health [4,24,25,26,27]. “The concept of worker well-being emphasizes quality of life and is driven by the relationship between individual worker safety and health and factors both at and outside the workplace, as well as a desire for workers to thrive and achieve their full potential [24,25,26,27]”. “Well-being integrates but goes beyond the traditional OSH goal of protecting workers from occupational hazards, to include preventing illness and distress and promoting worker health [4],” and competence development [4,25,26,27,28,29,30,31,32,33]. Well-being has physical, mental, and spiritual aspects [14,25,29,31]. For the framework to be useful and effective, it needs to be underpinned by core values. While such values are implicit in many of the papers cited to support the framework, there is a need to assert that core values, such as trust, interconnectedness, participation, justice, responsibility, growth, and resilience are necessary components of decent work and framework for the OSH field to address the future of work [29].

There is a need for a comprehensive framework that brings into focus for the OSH community issues related to the future of decent work. Such a framework is generally lacking. Earlier efforts were not as wide ranging or focused on OSH aspects and decent work as what is provided here [6,8,14,19,32].

## 2. Pillars and Determinants of Decent Work

The UN, through the ILO, has identified four major objectives that are the pillars of decent work. These include employment creation, social protection, rights of workers, and social dialogue [3]. Each of these has OSH aspects. For example, employment creation is a function of the level of worker training, the degree of precariousness, underemployment or even unemployment, and the adverse health effects arising from these. It is not only about employment as such, but also about high-quality jobs and jobs that meet the criterion of human dignity. Additionally, technological displacement and the degree of anxiety and stress that it causes are included here. Social protection is the integration of OSH practices and research to ensure safe and healthy working conditions, adequate rest, and access to benefits. Rights to work include the ability to bargain for wages and safe conditions, the ability to refuse unsafe work, and the reliance on employers to provide safe and healthy workplaces. Social dialogue is the right to exercise workplace democracy and have input on issues that might affect worker health and safety. These pillars, while defining ILO objectives to promote decent work, also represent economic, social, and political factors that drive, or in their absence, inhibit the achievement of decent work. They overlap and, in some cases, conflict, but they offer a useful normative foundation to improve the world of work.

The determinants of decent work are factors that positively or negatively influence the realization of decent work and raise issues that the OSH field will need to confront. The determinants used in the framework were selected by the authors for their significance and impact, and build on the literature that indicates their importance [1,8,9,11,14,15,16,17,22,26,27,29]. They include new and emerging hazards and risks, demographics (aging and gender), globalization, informal work, migration, pandemics, climate change, and OSH policies. Clearly, other factors, such as war or global conflicts, are also determinants of decent work, but they are not included here.

The identification of the OSH aspects of decent work can be accomplished by using the framework matrix in Figure 1, which shows the four pillars and includes the determinants of decent work. The challenge is to identify OSH issues and needed actions at each intersection in the matrix. In this article, we review the scientific literature and begin to address that challenge. Figure 1 is not an exhaustive list of determinants, but it represents major determinants that have OSH implications, as described here. This framework, named in the title of Figure 1, was developed for the OSH community to use as a conceptional staging area to consider OSH issues and actions needed to bring about decent work. There is no other framework in the literature, as far as we know, that covers such a broad range of determinants and their interactions with the pillars of decent work.

## 3. Identification of OSH Aspects of Decent Work

The OSH aspects of decent work can be determined through or the elaboration of the determinants of decent work, as discussed in this section. These OSH aspects can also be considered in how they align with the pillars of decent work.

### 3.1. New Technologies and New Forms of Work

In addition to addressing ongoing hazards, there is a need to identify the emergence of new occupational hazards [19,32]. Therefore, from this perspective, we will discuss some examples of new challenges imposed on OSH management systems by the constant changes and technological advances in the world of work to provide useful indications to adequately deal with these issues.

In doing so it is important to remember that the profile of risk factors that may affect health, safety, and well-being at work stem from complex relationships between the wider socio-economic context, employment, and working conditions. Furthermore, new forms of work organization and the expansion of the service-based economy have also resulted in new and emerging risks affecting workers, organizations, and society. An “emerging OSH risk” is described as a new and increasing occupational risk [32].

#### 3.1.1. New Hazards from Innovative Products and Substances: The Engineered Nanomaterials

Currently, thousands of products containing engineered nanomaterials (ENMs) are produced and marketed worldwide [34,35]. In parallel, human exposure to ENMs has increased in recent years, particularly for workers engaged in the manufacturing, handling, or use of ENMs [35,36,37]. The potential impact of ENMs on human health is a matter of great concern. ENMs can be absorbed through different exposure routes and are able to reach several organs and systems [38,39].

Experimental studies have shown that ENMs might induce several morphological and functional modifications, causing adverse effects at the respiratory, nervous, cardiovascular, immune, renal, and endocrine levels [39,40,41,42,43,44]. These effects occur through different molecular action mechanisms, involving oxidative stress, inflammation, or apoptosis, for example [44,45,46,47]. Nevertheless, a systematic and comprehensive risk assessment of these xenobiotics is still lacking, and different important shortcomings and gaps are yet to be filled.

For example, ENMs cannot be considered as a uniform group of substances; consequently, the adverse effects detected for a specific ENM cannot be generalized to other ENMs. Furthermore, limited data are available on the occupational exposure assessment of ENMs regarding both the environmental monitoring of personal workers’ exposure and biological monitoring [44,48,49,50]. Therefore, in concert with the implementation of an ENM risk assessment and management process that would be effective in improving exposed workers’ health and safety, there is a need for further studies in order to obtain more high-quality data. Additionally, there should be consideration of safety-by-design approaches so that hazards of new ENMS could be prevented or controlled.

Meanwhile, ENMs are being used in what are called “advanced materials”, in which the ENMs import new or improved properties that may be triggered by environmental conditions. ENMs are also used in the additive manufacturing (often called 3-D printing) of industrial materials and consumer products. These printers emit particles, volatile organic compounds, and other substances that could be hazardous to workers [37].

#### 3.1.2. New Hazards from Innovative Production Processes and Work Organization: Industry 4.0

The term *Industry 4.0*, first used in Germany in 2011, indicates an innovative conceptual idea of economic policy that is set on the implementation and integration of novel high-tech strategies and tools in the production system in order to “*digitally connect everything in and around a manufacturing operation in a highly integrated value chain*” [51]. Fundamentally, this phenomenon—which is characterized by greater automation and computerization and aims to transform industrial manufacturing processes in order to make them more efficient and flexible in response to the ever-changing market demand [52]—is already a concrete reality. In addition, it is likely to become prominent in most manufacturing processes globally over the next 5 years [52,53,54,55].

It is expected that the conditions depicted in the *Industry 4.0* paradigm will change work organization and production processes, modifying the method of performing job tasks and activities [54]. The main OSH issue is to identify the potential consequences and to understand if these can modify the exposure conditions of workers by introducing new hazards or increasing the extent of risk factors that already exist [54,55].

Available evidence suggests that Industry 4.0 could be associated with several issues and concerns, especially involving the increase in psychosocial effects or the increase in work-related injuries or illnesses related to engineering and human errors or mistakes in programming and interfacing with automated devices [48,56]. A big issue is to what extent will Industry 4.0 provide higher-quality jobs (skill discretion, task variety, and decision latitude) and lower-quality jobs [57,58,59]. Therefore, further studies should investigate at all levels of production any emergent occupational hazards or risks related to Industry 4.0 to identify adequate preventive and protective countermeasures, thus making the use of these new technologies safer. A promising perspective is the concept of Industry 5.0, as used by the European Commission and others, indicating human-centric and socio-centric (sustainable and resilient) perspectives [60].

#### 3.1.3. Emerging Physical Risks

One of the key emerging physical risks is sedentary work associated with high levels of physical inactivity [61,62,63,64]. Sedentary work and prolonged standing at work have increased due to the rising use of computers and automated systems at work. Work demands have been commonly cited as causes of physical inactivity [61], which has been found to be associated with increased health risks, such as coronary heart disease, type II diabetes, and certain types of cancers and psychological disorders (depression and anxiety) [62,63]. Physical inactivity is also linked to obesity, which might lead to back pain, high blood pressure, cardiovascular disorders, and diabetes [62,64].

Sedentary jobs are also associated with more musculoskeletal complaints and disorders, e.g., in the neck and shoulder, and upper and lower back, which are associated with increased sick leave and lower work disability [65,66,67,68]. Since the COVID-19 pandemic, increased teleworking, usually from home, has raised further concerns in relation to the suitability of equipment and work stations that are associated with ergonomic risks and musculoskeletal disorders (MSDs) [69].

An EU-OSHA report on the future of working in a virtual environment [69] highlights issues associated with the use of smart equipment and devices, such as virtual reality (VR) headsets, which can result in eye strain, repetitive strain injury, increased cognitive load, decreased situational awareness, physical disorientation, and motion or cyber sickness that can result in accidents [69]. Working with robots through VR interfaces and avatars can also lead to more cognitive load and technostress, especially where the robot controls the pace of work and outperforms the worker [70]. In warehouses, these new technologies can increase the number of picks per shift without the need for more employees, and they further simplify work tasks, thereby reducing the needed training time. Hence, we see a mostly digital Taylorism scenario in order picking [71].

Finally, it is important to remember that different types of risk interact to result in various outcomes. For example, biomechanical or ergonomic risks interact with psychosocial risks to generate MSDs [72,73,74,75].

#### 3.1.4. Emerging Psychosocial Risks

Many workers are no longer regular employees in single or fairly unchanged organizations [76,77]. There has been an increase in subcontracting, but also phenomena such as zero-hour contracts and the gig economy. As a result, new challenges have emerged, such as defining responsibility for worker health and safety and clearly understanding the impact of precarious jobs on employee health and well-being. Unstable income generation and job insecurity have been found to be related to mental health problems [78], and also associated with worrying about work availability [79]. Job insecurity has been found to be associated with reduced well-being (psychological distress, anxiety, depression, and burnout), reduced job satisfaction (e.g., withdrawal from the job and the organization), and increased psychosomatic complaints as well as physical strains (e.g., [80]). Precarious employment has also been found to be associated with inadequate training, carrying out less attractive tasks, having less influence on decisions made at work, and less fair rewards [77,79].

In the modern workplace, workers have to deal with high quantitative (such as high speed; no time to finish work in regular working hours), qualitative (such as increased complexity of tasks), and emotional demands (for example, in customer service jobs) [64,81]. The widespread use of Information and Communication Technology (ICT) has led to work intensification while, as discussed earlier, mechanization, automation, and computerization have led to increased interaction with, or even replacement of human activities by, machines. With the increased use of robots and cobots (collaborative robots) at work, cognitive load, changes to physical working conditions, and safety must be carefully managed [69,82]. For example, teleworkers may feel isolated, lacking support and career progression (e.g., [83,84]). Smarter device use has made availability for work easier, but has also resulted in a blurring of boundaries between work and private life. However, there is limited knowledge on the effective management of remote work [85]. With the blurring of boundaries between work and private life, workers may work longer hours, have difficulty disengaging from work, and feel both physically and emotionally exhausted, especially if they lack experience with virtual work and lack support [15]. Indeed, there is evidence that, in certain occupations, workers may resort to the use of performance-enhancing drugs, especially where they work long hours, their performance is strictly monitored by algorithms, or where they can obtain them online [86].

The Introduction of faster data processing, algorithmic management, and audible command technologies has already started to result in a faster pace of work and, in many cases, less control and autonomy of workers over their work [86]. Furthermore, algorithmic management of work and workers, monitoring technologies, as well as the Internet of Things and Big Data may lead to cybersecurity issues and data protection issues, ethical issues, and information inequality with regard to OSH [86,87]. Viale Pereira et al. [88] discussed the threat facing vulnerable groups from the challenges of digitalization and the possible bias in algorithms or in data, “*where AI [artificial intelligence] applications may replicate unwanted human behaviors such as manipulation, prejudice, and discrimination*”.

Finally, issues such as emotional demands, including harassment or bullying/cyberbullying and violence, will increase [64,69]. This could have serious detrimental effects, since the proportion of workers reporting symptoms such as sleeping problems, anxiety, and irritability is nearly four times greater among those who have experienced violence, bullying, and harassment than among those who have not [64].

Overall, psychosocial risks and work-related stress are expected to increase in the future, with a faster pace of work and less control over work [89,90], especially if this is machine-dictated. Changes in technology are expected to bring about frequent changes in work processes, as well as increased job insecurity and more frequent job changes [86]. Additionally, remote virtual work, mostly from home, may increase feelings of isolation and loneliness, and associated mental health problems, despite technological advancements supporting better communication [15,89].

In summary, a crisis of mental health in the work environment has accelerated in the past few decades and is likely to affect the future of work [91]. As noted by Dollard and Nesser [91], “*Work stress and mental health issues arise when work lacks meaning and job demands exceed the resources workers have to manage them*”. The core of this crisis is not primarily technological innovations, but rather how the work is organized. Psychosocial determinants, such as the level of control over work, work autonomy, work pressure, power imbalances, bullying, profound lack of meaning, alienation, and dehumanization, have been linked to work stress, bullying, burnout, physical health problems, and death [90,92,93].

To address these psychosocial issues, the OSH field should take a broader perspective than just focusing on individual workers. Such a perspective can illuminate situations in an organization that give rise to prevailing working conditions. Dollard and Karasek [94] proposed a “healthy conducive production” model in which the goals of production are considered as important as the psychological health of workers. “*If management is concerned about the balance of production goals and the psychological health of workers*,” they note, then working conditions will also be balanced [94]. The philosophy, values, and actions of management can be seen as the “psychosocial safety climate” [94]. The psychosocial safety climate can be measured and has benchmarks that predict future job strain and worker depression, and its use could help to improve and protect worker mental health through the provision of “decent work” [95].

A recent regulatory impact statement of proposed progressive changes to OSH regulations by the Australian Victorian Government to improve mental health in the workplace proposed that psychological safety climate ratings (organizational and jurisdictional) would significantly improve as a result of complying with new regulations. The regulations specify the development of a prevention plan, risk management, and reporting requirements. They also provide an evidence-based mechanism to model feasible benefits, including improved mental health and avoided costs [96].

### 3.2. Demographics

#### 3.2.1. Aging

Aging is an inherent factor that affects the achievement of decent work. The workforce of the world is aging in both developed and developing countries. Individuals may be working longer because of personal motivation and financial necessity. As working lives increase in duration, workers will have to adapt advancing technologies and technological requirements to more frequent changes. This will require training and re-skilling [97,98].

Aging and organizational factors affect workability across the lifespan and throughout one’s work career. Mid-career workers—those 45 years of age and older—may face bias that makes it difficult to obtain new employment. Effective measures are needed in order to support the voluntary prolongation of active working life [97,99,100,101,102].

New and young workers also face issues that can be barriers to decent work. As ILO concluded: “Youth employment is not just about jobs; youth employment can be decent only if it incorporates the other three dimensions of decent work as well: rights, protection, voice and representation….The age-specific difficulties that young women and men face in making the transition from school to work include: lack of employment experience; strict labour market regulations; mismatch between youth skills and aspirations and labour market demand and realities; constraints on self-employment and entrepreneurship development; and lack of organization and voice, meaning that youths have fewer channels through which to make their concerns or needs heard [103]”.

#### 3.2.2. Gender

Although workforce participation by women during the COVID-19 pandemic has decreased, the future of work will be characterized by increasing participation in the workforce by women [104,105]. Women often face different treatment in the workplace, compared with men. Few studies have characterized gender differences across occupations and industries [106,107,108,109]. In most sectors, compared with men, women perform more routine cognitive tasks, which are the most prone to automation [110,111]. Consequently, women face a higher risk of job displacement than men. Additionally, women face differential treatment in the workplace due to gender discrimination and stereotyping [99,111].

In contrast, there are needs to identify how the physical and psychological risk exposures of men and women workers differ; for instance, one should look at the differences in their societal roles, expectations, responsibilities, biological characteristics, and employment patterns [111,112,113]. In fact, the ILO has identified 10 recommendations for gender-sensitive OSH practice [114]. Although researchers report sex/gender differences in health outcomes of workers, in-depth assessment of these differences is lacking. In addition, the results are almost never discussed and hypotheses are seldom formulated about any observed differences between sexes/genders. Moreover, beyond the binary, gender diversity should be recognized in all OSH policies and practices.

### 3.3. Globalization

Globalization is the enhanced integration, interconnectedness, and interdependence of peoples, human societies, and countries, characterized by a global and fast flow of goods and services, trade and finance, and ideas and investments [115]. It has a long history, but became quite prominent and impactful in the later years of the 20th century. Globalization is not the result of the advent of innovative and revolutionary technological advances; rather, it is a state-driven process in which new international rules become more prominent than national governments and policies [116,117,118,119]. Theoretically, this globalization condition should ensure countries’ significant growth and economic progress, but in practice, the distinctive features of globalization have increased the gap between industrialized economies and developing or low-income countries [118,119]. Therefore, clearly, the phenomenon of globalization can significantly impact work and working conditions.

#### 3.3.1. Impact of Globalization on Work

The modifications of work caused by globalization are numerous, but cannot be generalized to all countries, since the changes induced by this process are partly dependent on different national variables, such as the level of industrialization, political/economic orientation, labor legislation, and welfare work [115,119,120,121]. However, globalization has had a strong impact on work everywhere, so profound as to change the nature of the work itself. For instance, it has made work more flexible in developed countries and progressively informal in most emerging or developing economies, where remuneration and security are unequally distributed [117]. Over the past 25 years, the labor force participation rate has gradually decreased (by 0.1–0.2% per year), reaching a value of 61% of the world’s working-age population in 2018 [122]. In the near future (at least until 2023), labor force participation rates are expected to further decline by 2% in upper-middle-income countries and by 1% in high-income countries [122].

Globally, the gender gap for workforce participation is closing, but slowly. In 2021, the World Economic Forum (WEF) reported that progress has stagnated and pre-existing gender gaps, before the COVID-19 pandemic, have amplified differences between men and women [123]. “*Almost 80% of men aged 15–66 are in the labor forces versus 58.6% of women of the same age group* [123]”. Using the Global Gender Gap Index, WEF found that 58% of the gender gap had closed. This is a marginal improvement since the 2020 edition of the report, and “… *as a result we estimate that it will take another 267.6 years to close.*” “*Educational attainment is the subindex with the smallest global gender gap and relatively low variation: 121 countries have closed at least 95% of their educational gender gap and 64 countries (more than one-third of the sample) already have achieved 99.5% gender parity* [123]”.

#### 3.3.2. Impact of Globalization on Working Conditions and Workers’ Health

Precariousness of work is one of the main and most serious consequences of globalization [78,117,118,124,125,126]. However, in developed countries, precarious work usually materializes in the contractualization of temporary jobs, whereas in developing countries, job insecurity takes the form of informal employment [118,124].

Globalization also has affected production processes significantly, creating global production chains characterized by special production techniques [115]. Consequently, the synergic and coordinated action of this innovative production system and of unprecedented progress in information and communication technologies has substantially changed the organization of work, leading to a shift of employment from industry to services [118,127,128,129]. However, this paradigmatic change in work organization from “blue collar” to “white collar” has resulted in benefits and better working conditions almost exclusively for workers in developed countries [130,131]. As a result of the ban on the use or production of certain specific hazardous substances (such as asbestos and pesticides), multinational corporations have shifted the production of these materials to less-industrialized countries, where public opinion about OSH issues is usually less pressing and/or the regulatory framework is more permissive [132].

Consequently, most workers in less-developed nations are engaged in low-quality jobs, exposed to obsolete technologies and work processes, such as handling, using, or producing hazardous substances [130,132]. Conversely, in developed countries, these changes in work organization, together with social, economic, and political reasons, have also caused an important increase in psychosocial risk factors that, in turn, severely affect workers’ psychological health [133,134].

A paradigmatic example of the aforementioned changes in developed countries is provided by the progressively increasing application of the Industry 4.0 framework [48]. For instance, Industry 4.0 could make work safer and healthier through the use of wearable monitoring technologies that continually check worker well-being or alert them to extreme heat, toxic gases, and harmful chemical substances. Furthermore, the use of robots in carrying out different industrial tasks, such as painting, welding, and assembling or hazardous operations in disaster areas, is expected to reduce exposure to chemicals and manual handling of heavy loads, thus reducing the related diseases and injuries [48]. However, despite these apparent benefits, psychological risk factors are likely to become more evident and important than physical ones, since workers in this organizational model experience increased mental overload, anxiety, greater responsibilities, lack of privacy from wearable sensors and loss of autonomy, as well as significantly reduced inter-human contact [48,90].

Therefore, there is an urgent need to carry out further research to globally identify the impact of globalization on work, working conditions, and workers’ health. Indeed, these data are fundamental to designing, developing, implementing, and disseminating appropriate strategies and countermeasures aimed at adequately tackling the possible negative effects of globalization. It will be important to define universally acknowledged standards that should be applied worldwide in order to provide equivalent worker health and safety across all nations, irrespective of their level of development [132].

### 3.4. Informal Work

Informal work is an often unrecognized part of the global economy, although it can bring many adverse consequences for workers and societies. Informality is characterized by work agreements without government-regulated conditions (e.g., OSH requirements), and informal workers can be self-employed or sub-contracted in such a way that the companies do not have to pay taxes [135]. This practice usually becomes even more common in times of economic crisis, when companies tend to decrease their workforce, increasing the number of unemployed people [136]. Workers, in turn, tend to seek alternatives to supplement their salaries or compensate for insufficient family income. Mostly because of the lack of opportunity, a portion of the population starts to accept occupations without an employment relationship, giving up benefits that are rightfully theirs [137,138].

A report published in 2018 by the ILO shows that 61% of the world’s worker population, approximately 2 billion people, works informally. The high prevalence of informality is usually associated with low levels of education and with poverty. Most of these informal workers (93%) live in emerging and developing countries, particularly in Africa, Asia, and Latin America [139].

Worldwide, the informal economy is heterogeneous and highly segmented by the economic sector, workplace, social group, and gender [136]. Informal work might include a part-time job for extra money, street trade, a small business without registration, and improvised work in homes. All these forms offer little or no training, guidance, or supervision; irregular working hours; improvised and precarious workplaces; fewer opportunities for employees to file complaints; and sometimes exploitation of child labor [139]. Informal work situations can promote an increased likelihood of occupational accidents and hazardous exposures, culminating in diseases and/or incapacities.

Some companies in the formal industrial economy, seeking to reduce costs, have adopted the practice of incorporating informal outsourced activities, as well as formal contractions as part of the productive process. In this case, low-skilled labor performs the industrial production steps informally. This informal work scenario includes precarious working conditions where compliance with legislation to protect workers against chemical exposure and other hazards tends not to occur. The workers do not have access to training to adequately handle chemical components and physical hazards at work, do not use personal protective equipment, and do not operate in an environment with adequate occupational hygiene [135,137,138]. Therefore, workers are potentially exposed to hazards and occupational levels of chemicals with toxic potential, in some cases near or within their homes, which may also affect their families. This raises concerns about the risks to workers and their families’ health and the respective environmental impacts, making it clear that research in this area needs to consider hazards and various sources of exposure influencing health outcomes.

More than chemical exposures, physical and biological hazards may be involved in informal work settings. For example, heat exposure significantly contributes to work-related injuries in small companies that do not have complete equipment, regulations, and training courses [140,141]. Occupational exposure to solar and artificial radiation should also be considered as a factor that affects workers’ health [142]. The informal work environment lends greater vulnerability to experiencing psychological distress and developing mental illness due to a lack of support and social isolation, as well as physical illness due to increased accidents and occupational diseases [143,144].

Commonly, informal workers are subjected to long working hours, shift changes, pressure for productivity, and monotonous work that may lead to sleep deprivation, excessive stress, and fatigue, and may contribute to a decrease in cognitive performance [145]. A higher risk of suicide among agricultural workers may be related to job instability and informality, added to the exposure to pesticides that affects the central and endocrine systems [146,147]. Child labor remains a public health concern, being associated with adverse health outcomes and increasing the risk of unemployment, exploitation, and marginalization [148,149]. Worker exposure to toxic metals could also be associated with biological hazards, including major susceptibility to severe respiratory diseases, such as influenza, pneumonia, and COVID-19 [150].

Scientific studies are needed to assess what chemical exposures these workers have due to informal occupational activities. Ferreira et al. [151] observed high levels of potentially toxic elements in air samples collected from the breathing zones of outsourced, informal, and household workers engaged in jewelry welding activities. The air samples collected inside the homes had concentrations of manganese, nickel, zinc, cadmium, and lead above the environmental limits stated by the Agency for Toxic Substances and Disease Registry (ATSDR). In addition, the concentrations of nickel, copper, zinc, cadmium, and lead even exceeded the permissible occupational exposure limits enforced by the Occupational Safety and Health Administration (OSHA). This same study observed inadequate working conditions within the homes and higher levels of lead in the blood of informal workers than in control individuals.

Informal jewelry production also increases the families’ exposure to cadmium in the workers’ household environment, as indicated by a higher level of cadmium in workers’ urine [152]. The environment can also be affected by irregular and illegal discharge of chemical residues arising from informal home-based work with jewelry [152]. Young people and adolescents involved in informal jewelry work tend to sleep less and to lose leisure time and study hours [148].

Pexe et al. [153] observed that hairdressers are occupationally exposed to formaldehyde at high concentrations; many of them do not have formal employment and do not receive professional technical training, including awareness of biosafety procedures. Mathee et al. [154] reported the risks of exposure to lead for workers and family members in the domestic environment in informal cottage industries, such as subsistence fishing, artisanal cookware, dismantling of batteries, and informal mining. Therefore, chemical exposure in an informal and uncontrolled environment should be an important factor to be considered in research on its effect on health.

The inclusion of all exposures suffered by informal workers, including internal and external factors, must be considered for assessing their impact on health. The exposome—all human exposures from conception to death—could, in the future, offer a cutting-edge approach to evaluating occupational and environmental exposures that cause chronic diseases [155,156,157,158,159]. Targeted and untargeted exposomic biomonitoring of trace elements and the toxic substances in biological fluids is of fundamental significance for human health assessment.

In the current scenario imposed by the COVID-19 pandemic, the cessation of economic activity via lockdowns and the reduction of movement have increased the unemployment rates around the world [160,161]. Large numbers of workers have been made redundant or placed on a temporary leave of absence, sometimes without pay. Inevitably, this significantly impacts individuals’ incomes and their ability to pay for rent and other expenses [162]. Such financial fragility may force workers to accept informal labor relationships, in turn losing their work rights and exposing them to health risks and financial insecurity.

Informal work may lead to hot spots of COVID-19, since the daily work makes it difficult to apply social distancing and quarantine measures [163]. Usually, these workers need to use public transportation and have reduced access to healthcare and personal protective equipment. Disparities in health and socioeconomic factors can be reflected in the unequal distribution of the COVID-19 disease burden, increasing the rates of infection and mortality in vulnerable populations [161,163].

Women in Informal Employment Globalizing and Organizing (WIEGO) develops projects focusing on the informal economy, including over 50 specialists based around the world, and seeks to increase the voice, visibility, and validity of poor workers, especially women [136]. The Informal Economy Monitoring Study (IEMS) led by WIEGO in 2012 evaluated the realities that informal workers face across 10 cities (Accra, Ghana; Ahmedabad, India; Bangkok, Thailand; Belo Horizonte, Brazil; Bogota, Colombia; Durban, South Africa; Lahore, Pakistan; Lima, Peru; Nakuru, Kenya; and Pune, India). The study found that urban informal workers are important to the city economy, but are unsupported by city policies and practices [164]. In addition, a project of the United Nations University World Institute for Development (UNU-WIDER), called Transforming Informal Work and Livelihoods, intends to understand the patterns and drivers of informality, providing knowledge for better policy-making concerning this sector [163].

The formulation of public policies to protect this sector is difficult because the informal economy is often underestimated or not recorded in national accounts. The diversity of informality also makes it difficult to formulate effective policies that benefit all workers [139]. Nevertheless, some policies—such as small enterprise development, poverty reduction, labor regulations, and simplification of business registration—should be implemented to transform the nature of employment [114,165].

For many families, although informal work guarantees their livelihood, it points to risks to workers’ health. Therefore, efforts to create new policies that will help this group are urgently needed. Occupational and public health policies should integrate economic and social factors to improve work conditions, thus also protecting public health and the environment.

### 3.5. Migrants

It was estimated that there were 281 million migrants in 2020, and of these, 160 million were migrant workers [166,167]. The number of migrants represents 3.5% of the global population, up from 2.8% in 2000. Much of the work that these migrants perform involves 3-D jobs—dirty, dangerous, and difficult. Recently, the term 4-D has been used to acknowledge that discrimination and other social determinants of health affect migrant workers [168].

International migration has been used by some destination countries as a tool to address the labor market shortage [169]. The absence of a global coordinated approach for migrant workers highlights a key deficiency in public, population, and global health, resulting in a lack of basic occupational health and dignifying workplaces [170].

#### 3.5.1. Europe

There are an estimated 33 million migrant workers in Europe, many coming from North Africa, Turkey, and areas in Asia [171]. The largest numbers of immigrants were in Germany (728,000), Spain (407,900), France (283,200), and Italy (247,500) in 2020 [172].

Germany reported the largest total number of immigrants (728,600) in 2020, followed by Spain (467,900), France (283,200), and Italy (247,500). Germany also reported the highest number of emigrants in 2020 (448,100), followed by Spain (248,600), Romania (186,800), and Poland (161,700) [172].

The WHO has prioritized health promotions in those regions, focusing mainly on communicable diseases, but there is growing awareness of non-communicable diseases, such as occupational health risks that also require targeted policy and culturally sensitive practices [173].

#### 3.5.2. North America

The United States and Canada receive many migrants, but they have different immigration policies and health and social support systems. Although economic development through labor market participation is articulated as a priority of Canadian immigration policy, new immigrant workers continue to find themselves in low-paying, precarious, and high-hazard workplaces without access to resources to protect themselves [174].

Around 40 million people in the United States were born in another country, accounting for about one-fifth of the world’s migrants, the largest proportion in the world. The number is projected to reach 80 million by 2065. Work is a key driver; 29 million immigrants are currently working or looking for work. In 2018, over 11 million were undocumented, with around 10 million engaged predominantly in 3-D jobs. Estimates show, that during the COVID-19 pandemic, approximately 5 million undocumented migrants kept the country running as healthcare and non-healthcare essential workers. Considering that over 40% of the U.S. foreign-born population came from Mexico, Central America, and South America, the biggest health disparity is evidenced mostly in Latinos, who are less likely to have access to healthcare than non-Latino whites [171,175,176].

#### 3.5.3. South America

Since 2015, a significant exodus of Venezuelan nationals has occurred, mainly to Latin American countries [177]. These people often traveled by foot, walking as far as Chile. This massive and sudden migratory flow of more than 6 million Venezuelans due to the political and financial crisis created many challenges for the region. Most migrants arrive in their host countries desperate for work and a source of income, which often results in immediate occupational vulnerability and precarious working conditions. The current Venezuelan crisis has also resulted in an unprecedented “brain drain”, which benefits the host countries. Millions of these workers often receive residence and work permits in host countries through ordinary and favorable extraordinary processes [177].

This phenomenon has resulted in millions of residence permits and work permits issued to Venezuelan nationals by ordinary and extraordinary migration-normative tools. Consequently, the Venezuelan migrant population has also benefitted from the approval of specific legislation in the region, allowing legal relocation and often providing access to healthcare and other benefits [177].

#### 3.5.4. Africa

The high mortality associated with silicosis and other lung diseases and the epidemics of tuberculosis and human immunodeficiency virus among migrant mine workers are prominent occupational health narratives in southern Africa [178]. Efforts in the past decade to provide compensation for over 200,000 miners who worked in South African mines during the apartheid era and suffered occupational lung disease included long-standing claims by many migrant workers from neighboring southern African countries [179]. This process provides useful and historical insight into the coercive migrant labor system, deficiencies in the public health system, and the social determinants of migrant workers’ illness.

#### 3.5.5. Asia

Six of the world’s ten largest workforces are in the Asian region, mostly engaged in small enterprises or in unregulated informal sectors. With limited employment opportunities in their own countries, many migrate as temporary workers and become a valuable source of income for their home countries. India is the world’s largest source of migrant workers and is simultaneously becoming a recipient nation, with increased regional migration to India. The South Asian countries and the Gulf States account for a large proportion of migrant workers in precarious jobs, attracting unskilled migrant labor over the past three decades, mostly service sector workers from East Asia and construction workers from Southern Asia. It is common that working conditions in severe climates of heat and humidity create a daily risk for the millions of migrant workers in that region.

Regular reports of work-related deaths and injuries of migrant workers in sectors such as construction and agriculture, in addition to precarious low-wage working conditions for female domestic workers in the service and hospitality sectors, have lately received mass media attention. Nevertheless, because of the limited research output on the occupational health of migrant workers, especially females, in regions with low research contributions, policymakers have overlooked the problem [180,181].

Half the working population in the world is unemployed or underemployed [182,183]. The UN, WHO, and ILO need to consider how their programs can be delivered more effectively [184] and acknowledge the substantial contributions by migrant workers to both host countries and their countries of origin.

### 3.6. Pandemics

Pandemics could have strong modifying effects on the future of work and the realization of decent work. The recent COVID-19 pandemic has illustrated how some types of work have become decentralized; however, “deskless” workers still had to perform their jobs and were more likely to be exposed to the virus. The economic crisis resulting from COVID-19 has hit some job categories harder than others, making workers (such as those in temporary jobs, informal employment, or diverse work arrangements) who already had limited means and protection even more vulnerable [185].

The COVID-19 pandemic has exacerbated the negative unemployment effects that globalization had already partly induced. However, in this regard, it is worth noting that, although the increase in unemployment due to globalization was a natural consequence of this phenomenon, the unemployment related to the pandemic is instead a consequence of the weaknesses of globalization. The efficiency of the globalization model is based on a complex web of supply chains connecting manufacturers around the world. However, when both the effects of the COVID-19 pandemic and forced closures of industries compromised the production capacity of component manufacturers, numerous producers of related consumer goods were forced to shut their factories, thus leaving workers at home and causing a significant rise in unemployment [69]. Then, the global value chains, which have always represented the strength of globalization, suddenly became its weakness.

With regard to working conditions and workers’ health, it should be considered that, in some work sectors, particularly those working with computers, the COVID-19 pandemic is profoundly changing working conditions by promoting the spread and implementation of different forms of flexible work (smart working, teleworking, and so on). Indeed, as a consequence of lockdowns or stay-at-home measures adopted by governments, a significant proportion of workers have been assigned to work remotely at home. In this way, where it was possible to implement these flexible working methods, health needs (limiting the movement of people to reduce contagion) were combined with economic ones (ensuring business continuity).

Nevertheless, the sudden change from office- to home-based working might have serious consequences for OSH, especially regarding psychosocial risks and ergonomics [186]. For these reasons, in the near future, it will be important to pay more attention to these aspects, including technostress and technology overload, prolonged sedentariness, ergonomic adequacy of home furniture, isolation, and the work–life interface [186].

In the two decades before the COVID-19 pandemic, the world had already faced two other outbreaks of novel coronavirus disease: severe acute respiratory syndrome (SARS) in 2002–2003 (8098 confirmed cases and 774 deaths) and Middle East respiratory syndrome (MERS) in 2012–2019 (2502 confirmed cases and 861 deaths) [187]. Therefore, considering both the frequency of these events and their potential disastrous effects, there is a need for innovative tools and strategies that allow us to counteract these events properly and timely, in the OSH field and elsewhere [188].

### 3.7. Climate Change

Increasingly, climate change will have an impact on work, workers, and workplaces. Workers in all sectors—but particularly outdoor workers—will be affected [189]. However, past and recent studies indicate that indoor workers are also at risk of adverse effects from climate change [190,191]. Climate change can affect the quality of work and the potential for decent work, and particularly impacts workers who are disproportionally affected by socioeconomic disparities. Climate change affects workers more than the general population, because workers often have exposures of longer duration and greater intensity. Climate change also significantly interacts with chronic diseases to cause or exacerbate them [189]. A critical effect of climate change is increased ambient temperature, which is likely to result in an increase in the number of workers dying from heat stress, as well as those made ill from it. Heat stress also has been shown to lead to decreased productivity [192]. Temperatures above 24 °C to 26 °C are associated with decreased worker productivity [192].

In addition to heat, other hazards related to climate change have been identified. These include air pollution, ultraviolet radiation, extreme weather, and expanded habitats. Additionally, hazards arise with industry transitions from old to new technologies and from the built environment [123,189]. With all these hazards and their attendant scenarios, mental health stressors occur and can have adverse effects.

The hazards of climate change can also cause extensive displacement of workers and work, resulting in inter-country and intra-country migration of workers, leading to unemployment and underemployment. Such migration is expected to lead to global productivity loss and the displacement of millions of workers [189,193,194]. The working capacity of heat-exposed workers, on average, is expected to decrease globally throughout 2021 [194,195]. Climate change will have effects on economics, social affairs, and OSH [124,189].

There are potential synergies for decent work and climate change policies. Decent work, green jobs, and sustainable development are overlapping objectives [124,194,195]. The impact of climate change varies across employment sectors and geographic regions and, in many cases, exacerbates inequality and the lack of potential for decent work.

There is a need for appropriate policies, technological investments, and behavioral change to cope with climate change [124,192,193,194,195,196,197,198]. At the working level, employers are not used to addressing climate hazards as part of their responsibilities to provide safe and healthy working conditions. New awareness and resultant control actions by employers are needed to address these hazards.

Climate change can be a source of job creation that may even outnumber the jobs that are displaced. Critical to such efforts is the need for these to be quality jobs in terms of wages, benefits, and working conditions [195,196].

Climate change policies can be considered as belonging to two types: mitigation and adaptation. Both of these can have an impact on workers. If mitigation is not timely or likely, then the focus must be on adaptation and resilience [196]. National and international policies are central to addressing the mitigation of climate change. As Dunne et al. [195] noted, positive outcomes of policies require “*cohesive sets of country-specific policies, articulating economic, environmental, sectorial and enterprise policies with social and labor policies*”.

The adverse effects and costs of climate change are likely to be disproportionate for certain communities, industries, and workers. Consequently, there is a need for efforts to minimize these effects [197]. Governments, employers, and workers should promote the development of transition plans for these groups at risk. Barrett [198] identified the elements of such a plan, which include providing a range of support programs for workers affected by climate change.

Information on the mental consequences of climate change for workers is increasing [196,197,198,199]. Additionally, another phenomenon has been labeled—“Eco anxiety” is a grief reaction affecting people about the state of the planet and the significant role humans have played in its gradual degradation [196,200]. Because climate change is a serious hazard for workers, it is important that it be an integral component of OSH and decent work. This includes the incorporation of climate change in etiologic and intervention effectiveness research, and also in surveillance, risk assessment, and risk management [189].

### 3.8. OSH Policies

Policies can be proposed or adopted at the macro level (e.g., internationally or regionally), meso level (e.g., provincially or sectorally), or micro level (e.g., organizationally) [27]. Here, we consider policies addressing OSH and well-being at the first two levels. Policy instruments are typically differentiated as hard law or soft law [27]. Hard law implies legally binding obligations that are precise and delegate authority for interpreting and implementing the law [201]. Examples of hard law include statutes or regulations in national legal systems [202,203]. At the intergovernmental level, examples include legally binding treaties, conventions, and directives. In contrast, soft-law development relies on voluntary participation and consensus-based decision-making for action. Therefore, soft law is voluntary in nature, and examples include sectoral agreements and standards [27].

Ideally, policy approaches should complement each other, whether they are focused on public or occupational health issues, economic issues, social security, or sustainability [27]. However, it is widely acknowledged that such complementarity is rare due to the differing perspectives and priorities of policymakers [202,203]; this also applies to policies of relevance to health, safety, and well-being. Furthermore, the context in which policies are developed and implemented is important, and as a result, policy approaches vary across countries, whether developed or developing/newly industrializing [27].

These challenges include addressing employers and worker representatives related to independent workers, the balance between policy and regulation, the balance between business flexibility and worker autonomy, worker privacy and employer use of algorithmic monotony, the interaction of humans and robots, manufacturing worker well-being and productivity, and enhancing life-long workforce diversity perspective [204].

#### 3.8.1. Hard Law

Regulation at international, regional, and national levels is considered a significant driver of the promotion of health, safety, and well-being at work [205]. An example of international hard law are OSH-related conventions developed by the ILO, which seek to establish basic standards to ensure workers’ health and safety. While ILO conventions No. 155, No. 161, and No. 187 are recognized as the three key OSH conventions, there are several additional ILO conventions of relevance to health, safety, and well-being [27].

National legislation must conform to standards established in international regulation (e.g., if the country has ratified an ILO convention) and regional regulation (e.g., EU directives); however, there is large variation in the scope and coverage of national OSH legislation [206].

#### 3.8.2. Soft Law

Soft law instruments also exist at the international level. Examples include ILO recommendations, such as Recommendation 164 on Occupational Safety and Health (1981), which is directly relevant to the convention of the same title. Other examples are standards set by the International Organization for Standardization (ISO), such as ISO 45001, an OSH management standard that aims to promote a comprehensive approach, and ISO 45003 (2021): Occupational Health and Safety Management—Psychological Health and Safety at Work—Guidelines for Managing Psychosocial Risks [27].

Other soft law instruments are social partner agreements or guidance and tools available at a regional, national, or sectoral level. For example, European social partners (employers’ organizations and trade unions) have concluded a number of “voluntary” or “autonomous” agreements, including framework agreements on telework (2002), work-related stress (2004), harassment and violence at work (2007), inclusive labor markets (2010), and digitalization (2020) [27].

Finally, numerous examples of guidelines that address health, safety, and well-being have been developed at international, national, sectoral, and organizational levels. At a national level, guidelines are often complemented by tools that organizations can use in order to implement good practice as specified in the guidelines. A popular example is the Management Standards for Work-Related Stress, which were developed by the Health and Safety Executive (HSE) in the United Kingdom [207] and have been adapted by the National Institute for Insurance Against Accidents at Work (INAIL) in Italy [202,203] and the Health and Safety Authority in Ireland (Work Positive) (see, e.g., [202,203] for a discussion of the evaluation of the Italian approach). In the United States, the American National Standards Z10 management standard also provides a soft law approach [208].

#### 3.8.3. Finding Balance in Hard and Soft Law Approaches

The multifaceted nature of health, safety, and well-being at work has meant that various policy approaches, both binding (hard law) and voluntary (soft law), have been developed and implemented to address the issue. Jain et al. [27] discussed the pros and cons of the two approaches, and argued that hard law offers legitimacy, strong surveillance and enforcement mechanisms, and guaranteed resources that soft law often lacks. Furthermore, hard law has been found to be one of the most important motivators for organizations to engage with health, safety, and well-being [205,209,210]. However, there are also some drawbacks in solely promoting a hard law approach.

For example, in the EU and other developed countries, OSH regulation covers both traditional health risks (e.g., physical risks) and emerging risks (e.g., psychosocial risks). However, as the focus has moved towards the prevention of ill health, the regulatory approach has been found to be less effective [74,211] due to the lack of specific coverage of emerging risks and unclear terminology on them. Importantly, a hard law approach would be most effective in those countries where there is a more advanced framework to effectively translate policy into practice [27]. However, in those countries where OSH legislation does not meet international standards and is not appropriately enforced [212,213], relying on hard law alone might not be sufficient. Since the developing world is where a strong focus on OSH is needed the most, a strong argument remains for a supplementary strategy [27].

Another issue is that nations might choose not to use hard law instruments, such as not ratifying ILO OSH conventions, to avoid strict adherence to their specifications. Furthermore, in many countries, there is a desire to minimize the regulatory burden placed on organizations, especially small and medium enterprises (SMEs) [74], since this might be seen as hindering agility and economic growth. Another concern is organizations transferring hazardous operations and production to low-income countries in order to avoid jurisdictions with demanding legal environments, or even business lobbying for changes in OSH legislation [27]. Finally, “creative compliance” is also an issue, in which organizations adhere to legislation only superficially, but do not violate OSH legislation in a strict sense, and therefore cannot be prosecuted [27].

On the other hand, soft law approaches, which are based on consensus among a broad array of stakeholders, can provide: (1) timely actions when governments are deadlocked; (2) bottom-up initiatives with legitimacy, expertise, and resources to enforce new standards; and (3) efficient methods for direct civil society participation in global governance [27]. Soft law (e.g., codes of corporate social responsibility) has also been found to be generally more precise and user-friendly than hard law in relation to health, safety, and well-being [214].

Nevertheless, soft law may also lack the legitimacy and strong surveillance and enforcement mechanisms offered by hard law. Since many stakeholders are involved in its development, soft law may be perceived as promoting compromise standards that are less rigorous than those promoted by governments [27,204].

Both hard and soft law approaches are directly impacted by the context in which they are developed, the actors who participate in their development and their perception of OSH risks, and the process of the negotiation and implementation of these approaches. These issues affect the actions taken by governments, regional institutions, and organizations to manage health, safety, and well-being [27]. Finally, with new and emerging risks affecting work and the workforce, including risks associated with the COVID-19 pandemic, digitalization, and remote and virtual work, new challenges must be addressed in policymaking [81].

One key challenge is keeping pace with developments [81], since research and policy often lag behind changes in practice. For example, a lack of knowledge on newly emerging OSH risks complicates policymaking and enforcement. Another key consideration is the responsibility of employers and workers and social protection, given the rise of the independent worker that was discussed earlier, as well as ensuring that regulation does not hinder technological progress and negatively affect organizational and national competitiveness [81].

According to Leka [81], policy frameworks need to be re-examined in light of new and emerging risks, and a good balance must be found between hard and soft law. Existing regulation needs to be updated regularly and clarify OSH liabilities and responsibilities, as they relate to new systems and new ways of working [81,192,215]. Soft law instruments, such as standards and voluntary social partner agreements, can also continue to play an important role. A good example is the recent European Social Partners Framework Agreement on Digitalisation of 2020, which applies to all industrial sectors and covers work organization, working conditions, work content and skills, and work relations [81]. Considering sectoral differences in terms of telework and virtual work, sectoral approaches would hold great potential, as well as holistic policy models that adopt a lifelong perspective to working life with a strong focus on well-being [81,216,217].

Importantly, ethical issues need to be strongly considered, and the development of codes of conduct could prove helpful in addressing these [81,218,219]. New technology might indeed present opportunities for collective worker representation and bargaining, and for inspection through the use of Big Data and smart devices [69,219]. Finally, providing effective OSH services for virtual workers using new technology should be considered [81].

## 4. OSH-Related Components of the Pillars of Decent Work

The foundation for considering the OSH aspects of the determinants and pillars of decent work are discussed in Section 3. Table 1 shows the OSH aspects as they track with the four pillars of decent work. Clearly, many of these aspects are beyond the traditional focus of OSH, but they can be addressed with an expanded focus, as called for by various authors [4,8,9,11,12,13,14,19,220]. The OSH community needs to understand its role among the many specialties promoting well-being and look for connections between OSH and broader efforts to improve wages, job security, and worker voice. The OSH community may lend support to these larger efforts where these connections become visible. The framework, shown in Figure 1, serves as a tool for the OSH field to have an expanded focus leading to decent work.

Various authors have addressed the need for the OSH field to expand its focus to address issues that will arise and extend into the future [6,9,10,11,12,13,14,19,24,25,26,27,31,86,89,90,220]. Crossing many of these analyses are issues of transdisciplinarity and systems thinking [216,220,221,222,223]. The perspective on decent work must incorporate or expand systems thinking, strategic foresight, health equity, and transdisciplinary concepts in the OSH field [4,220,221,222,223,224,225]. In summary, the field must evolve forward to a more comprehensive, public health-oriented model of worker health [4,9,223,225]. It should use a salutogenic perspective as a means of achieving well-being, which is a prime characteristic of decent work [225].

More specifically, for decent work to be achieved for use by the OSH field, the concept needs to be operationalized. There is a need for a comprehensive definition that is grounded in a social justice ethos and explicitly captures the psychological aspects of working [24,226,227,228,229]. The OSH field needs to develop more of a psychosocial perspective [24,226,230]. Such a perceptive pertains to psychological and social health, but also, as the UN (2014) envisioned, “… *a state of well-being in which every individual realizes his or her own potential, can cope with the normal stress of life, can work productively and fruitfully and is able to make a contribution to her or his community* [228]”. Clearly, there is a strong evidence base that supports the relationship between work and psychological health [24,226,229,230]. While there is a continued need to focus on empirical research [231], there is value in “… *qualitative, discovery oriented research as a tool to unpack how people experience their working contexts*” and their perception of whether they have decent work [24]. Occupational health psychology and other social sciences (e.g., economics, sociology, anthropology, management science, and human relations) will increasingly be needed to optionally create conditions, that promote decent work and enable the OSH field to contribute to it [4,24,26,27,30,220,223,225,226,229,231].

The OSH aspects of the pillars of decent work related to workers and workforce should not be seen as a one-time assessment, but rather viewed over the entire working lives of workers [232,233,234]. A life course perspective helps to understand the cumulative effects of work, underemployment, unemployment, burnout, and blended work/retirement situations [232,234]. Decent work is a function of work that is secure over a working life. A life course perspective should be integrated into the consideration of decent work and applied to research, policy, and practice [226,232,233]. Decent work is also ultimately a function of good management, since employers and managers are responsible for resources and the organization of work and workplace well-being [31,93,234,235]. Foremost, the OSH community should increase its efforts to influence employers and managers.

The framework in this paper is underpinned by the growing view that the health and safety of workers is a human right. The principle of a safe and healthy working environment has been added to the ILO’s *Fundamental Principles and Rights at Work* [236].

Nevertheless, the framework may have inherent biases relative to developed countries. The question arises of how it can be applied outside such countries, where practices and conditions do not easily allow for control of hazards and risks and provisions of jobs for self-actualization. Achieving decent work is a global goal, but there is a need to realize that the application of the framework will be dependent on the state of development of countries and will not be achievable at the same time everywhere. International reflection on this differential and appropriate technical support will be one way to address it and move toward decent work for all.

## 5. Conclusions

This report elaborates a framework for anticipating the OSH features of decent work in the future. The framework identifies opportunities to consider OSH-related challenges that occur in a matrix at the intersection of the ILO pillars of decent work (on one axis) and the authors’ identification of important determinants of decent work (on the other axis). Most of the emphasis is on the determinant axis of the framework, but the novelty of this framework is that it provides a conceptual space to raise questions for the OSH community that arise at the intersection of determinants of decent work and ILO’s objectives (pillars) to achieve it. If the OSH field is to be prepared to address the future of decent work, it will need to broaden its focus to operational concepts of well-being and use of systems thinking, strategic foresight, and transdisciplinary and public health approaches. Additionally, holistic models will need to be realized that adopt a lifelong perspective to working life with a focus on well-being. By expanding its focus, the OSH field will be better prepared to make important contributions to decent work and the well-being of the workforce and the population.

## Figures and Tables

**Figure 1 ijerph-19-10842-f001:**
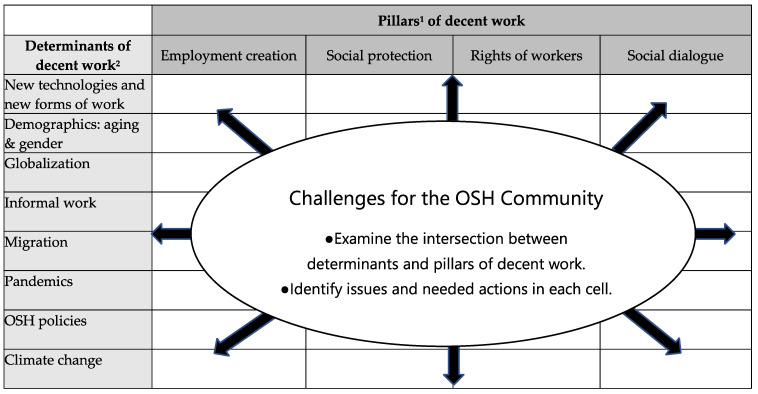
Occupational Safety and Health Staging Framework for Decent Work. ^1^ Pillars are strategic objectives of the ILO to promote decent work [3]. ^2^ Determinants are factors that positively or negatively influence the realization of decent work.

**Table 1 ijerph-19-10842-t001:** Occupational safety and health-related components of the pillars of decent work.

Employment Creation	Social Protection	Rights of Workers	Social Dialogue
➢ Technological displacement of jobs ➢ Retention of older workers ➢ Reskilling and upskilling of workers ➢ Development of quality work (task variety, skill discretion, and decision latitude) ➢ Good employment (security and equal pay)	➢ Sensitivity training to retain workers ➢ OSH ➢ Health insurance ➢ Disability protection ➢ Appropriate hours of work ➢ Work–life balance ➢ New hazards and risks	➢ Discrimination against older workers ➢ Collective bargaining ➢ Gender sensitivity to physical and psychosocial risks ➢ Psychosocial safety climate ➢ 4-D Jobs (Dirty, Dangerous, Demanding, and Discriminatory)	➢ Tripartite consultation ➢ Participatory research ➢ Employee requests for PPE ➢ Voluntary agreements ➢ Bottom-up models

## Data Availability

Not applicable.

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
