# Peer review of "Occupational Safety and Health Staging Framework for Decent Work"

_ijerph, 2022, doi:10.3390/ijerph191710842_

Round 1

Reviewer 1 Report

Overall considerations :

This paper deals with an important and relevant topic. ICOH is a recognized society and a reference in the field of Occuaptional Health and the team of authors appears as a guarantee for the quality of the paper content. It is well structured and clearly presented and contains a lot of useful references to document, support and illustrate the presented facts, comments, hypotheses, lack of knowledge  and potential consequences and impacts of some factors.

The choice of the determinants of decent work has been made by the authors for their significance and impact (page 3, line 110-111) which means that it is somewhat subjective and emphazises on specific factors and pays little or no attention to others. This could be underlined in the second paragraph of chapter 2 where the statement of this choice is made. In other words, it is not a comprehensive suvey of the determinants but  doesn't detract in any way from its worth. 

Positive aspects

The numerous elements presented in the chapter 3 (Identification of OSH aspect of decent work) are quite relevant and emphasize very important factors and needs for research. The chapter 4 (Regulations) is also helpful to understand the impact of hard and soft laws.

Needs of improvements (see also the attached file : Elements for review)

Workers' well-being is considered only from a psychological point of view limited to mental health and major factors such as the moral issues and the core values at work are ignored. Solidarity, for instance, is also one value that builds unity in teams, in companies and in communities. This lack of core values represents for me a serious weakness in this framework to identify relevant OSH aspects of decent work that has to be meanginful and totally fulfilling.

The evolution of managment and the emerging new models represent a very important factor to improve working conditions and cannot be ignored in the context of decent work. The roles of managers and persons responsible of human resource are determining and should also be emphasized.

Salutogenesis is the science focused on health (and not on diseases like medicine) contributes to understand which factors play a key role on the development of good health, including health at work. Although it is slowly emerging it has to be mentioned in this context. 

Table 1 that presents the matrix of pillars of decent work and the chosen determinants doesn't bring anything beside ligheting the text and should be removed or modified. It could be summarized in one sentence put in epigraph.

Table 2 is hard to read due to the non-alignment of the components of the pillars and should be improved.

Details

The first keyword is "Migrant workers". Does it mean that this is the most important topic relatd to decent work ?

Reference 33 does not concern nanoparticles

Reference 87 does not concern harassment (page 6, line 265)

Corporate social responsability and its related ISO reference is missing in chapter 4.2

Author Response

Reviewer 1

1. This paper deals with an important and relevant topic. ICOH is a recognized society and a reference in the field of Occupational Health and the team of authors appears as a guarantee for the quality of the paper content. It is well structured and clearly presented and contains a lot of useful references to document, support and illustrate the presented facts, comments, hypotheses, lack of knowledge and potential consequences and impacts of some factors.

Response

Thank you.

2. The choice of the determinants of decent work has been made by the authors for their significance and impact (page 3, line 110-111) which means that it is somewhat subjective and emphazises on specific factors and pays little or no attention to others. This could be underlined in the second paragraph of chapter 2 where the statement of this choice is made. In other words, it is not a comprehensive suvey of the determinants but  doesn't detract in any way from its worth. 

Response

We added to the rationale for selecting the determinants.

3.The numerous elements presented in the chapter 3 (Identification of OSH aspect of decent work) are quite relevant and emphasize very important factors and needs for research. The chapter 4 (Regulations) is also helpful to understand the impact of hard and soft laws.

Response

Agreed.

4.Workers' well-being is considered only from a psychological point of view limited to mental health and major factors such as the moral issues and the core values at work are ignored. Solidarity, for instance, is also one value that builds unity in teams, in companies and in communities. This lack of core values represents for me a serious weakness in this framework to identify relevant OSH aspects of decent work that has to be meanginful and totally fulfilling.

Response

We clarified that well-being is mental, physical and spiritual. 

5. The evolution of managment and the emerging new models represent a very important factor to improve working conditions and cannot be ignored in the context of decent work. The roles of managers and persons responsible of human resource are determining and should also be emphasized.

Response

The issue was added to section 5.

6. Salutogenesis is the science focused on health (and not on diseases like medicine) contributes to understand which factors play a key role on the development of good health, including health at work. Although it is slowly emerging it has to be mentioned in this context. 
Response

Salutogenesis is a major concept in reference 228, and it was made more explicit in the text.

7. Table 1 that presents the matrix of pillars of decent work and the chosen determinants doesn't bring anything beside ligheting the text and should be removed or modified. It could be summarized in one sentence put in epigraph.
Response

This table was modified and changed to a figure. Its utility is that it gives a visual depiction of the framework.

8. Table 2 is hard to read due to the non-alignment of the components of the pillars and should be improved.
Response

Table 2 has been fixed.

9. The first keyword is "Migrant workers". Does it mean that this is the most important topic related to decent work?
Response

We don’t believe that the order of keywords has any significance, however we have moved “migrant workers” from the first position.

10. Reference 33 does not concern nanoparticles
Response

Reference changed. 

11. Reference 87 does not concern harassment (page 6, line 265)
Response

Reference changed.

12. Corporate social responsibility and its related ISO reference is missing in chapter 4.2
Response

A reference (216) on CSR was already there, it was made more explicit in the text. Other references were not considered necessary.

Reviewer 2 Report

-On the whole, the introduction raises important and interesting questions; What is the OSH community going to do to promote decent work in the light of demographic, economic, social, technological and pandemic changes?

As anyone can see, these are large and highly ambitious questions, which inherently asks new questions: what is decent work? and what are these determinants (as used by the paper; demography, economic, techno etc.). 

-In the introduction I would like to see the paper relate itself to other frameworks more carefully. you cite earlier models and efforts but you do not nearly deep enough engage with what these older approaches do and why your framework is needed and contributes to the field. 

- also this improved positioning should translate into the main parts of the text and to the conclusion.

- also, I would like the methodological story of how this paper was born. to me it looks like a sort of oppinion paper as the basis for the framework is not built on a review or similar scientific practice. One may then ask oneself is this is the best way of constructing a framework. But I accept that you have to start somewhere.

- in the main text pieces. I do agree with the different important aspects or determinants of decent work, but I would urge the author towards a more humanocentric or philosophical engagement with the different aspects. this could lead to more elaborate discussions of the considerations and dilemmas in addressing each determinant.

- finally I think the authors should perform some amount of conceptualization of their framework: 1) it needs a name, 2) it needs a better visual representation at current neither table 1 or table 2 are suited for publication. finally, a third figure or table displaying the framework and its usefullness could be improved.

- small issues. In line 56. I would suggest using 'continuously' rather than 'increasingly'. the world of work has been changing over all the history of humanity.

line 198 two dots.

Author Response

Reviewer 2

  1. On the whole, the introduction raises important and interesting questions; What is the OSH community going to do to promote decent work in the light of demographic, economic, social, technological and pandemic changes?

Response

Yes, this is the type of question that the framework is supposed to raise. The paper just provides a framework to identify where such questions will arise so OSH practitioners can consider what they would do.

  1. As anyone can see, these are large and highly ambitious questions, which inherently asks new questions: what is decent work? and what are these determinants (as used by the paper; demography, economic, techno etc.). 

Response

Yes.

  1. In the introduction I would like to see the paper relate itself to other frameworks more carefully. you cite earlier models and efforts but you do not nearly deep enough engage with what these older approaches do and why your framework is needed and contributes to the field. 

Response

A statement on why the framework was needed was added.

  1. Also this improved positioning should translate into the main parts of the text and to the conclusion.

Response

Language was added to the conclusions to carry over the theme.

  1. Also, I would like the methodological story of how this paper was born. to me it looks like a sort of opinion paper as the basis for the framework is not built on a review or similar scientific practice. One may then ask oneself is this is the best way of constructing a framework. But I accept that you have to start somewhere.

Response

This paper was identified in the ICOH Future of Decent Work workgroup. The framework was developed through group interactions and the impetus for it was the changing world of work and the need for the OSH field to evolve to be effective. A note was added to the Acknowledgements.

  1. in the main text pieces. I do agree with the different important aspects or determinants of decent work, but I would urge the author towards a more humanocentric or philosophical engagement with the different aspects. this could lead to more elaborate discussions of the considerations and dilemmas in addressing each determinant.

Response

Language was added to emphasize the need for humanocentric engagement.

  1. finally I think the authors should perform some amount of conceptualization of their framework: 1) it needs a name, 2) it needs a better visual representation at current neither table 1 or table 2 are suited for publication. finally, a third figure or table displaying the framework and its usefulness could be improved.

Response

The framework has been named and made into a figure and Table 2 has been modified.

  1. small issues. In line 56. I would suggest using 'continuously' rather than 'increasingly'. the world of work has been changing over all the history of humanity.

Response

Change made.

  1. line 198 two dots.

Response

Unclear what this means, line 198 seems correct.

Author Response

Reviewer 3

  1. This manuscript offers a global framework for guiding occupational safety and health (OSH) research, practice, and policy in the future. There are many things to like about the manuscript, including its strong foundations in the International Labor Organization’s (ILO’s) framework for “decent work,” its attentiveness to ongoing and continuing trends that pose updated risks to worker well-being, and its attempts to move on beyond classical (OSH) of eliminating hazards/risks in the workplace alone. Each of these ideas and those unmentioned are meritorious and worthy of attention as OSH redefines its professional role and identity. Nevertheless, the manuscript ultimately falls short of achieving its objective of providing a helpful framework for guiding OSH into the future.

Response

We have made changes to the paper, based on the reviewers’ comments and believe it is more helpful for the OSH community.

  1. Core features of the framework’s conceptualization are flawed, raising questions about its ability to guide OSH into the future meaningfully. I noticed three specific problems in the proposed framework.

23a. The “Pillars of Decent Work” are presented as independent when the evidence suggests otherwise. For example, there is clear evidence that employment creation (one “pillar of decent work”) conflicts with social protection (and perhaps rights of workers) in practice, at least in developing economies (GümüÅŸ & Gülsün, 2020; Li et al., 2021). By contrast, as economies grow, there seems to be a “developmental effect” wherein greater prosperity manifests in more worker protection and reduced injury and fatality (see Palaz & Colak, 2017). If the authors want to use the concept of “decent work” as the ILO conceives it – presumably for policy purposes – the authors need to be transparent and grapple with the complex interrelations among the “pillars” to create actionable guidance for OSH.

Response

We state that “… determinants are factors that positively or negatively influence the realization of decent work.” The critical aspect is that they can influence as an effect modifier, or as indirect or direct causes. The examples that the reviewer raises illustrate the dilemmas OSH practitioners must face. Language to the effect has been added.

23b. The meaning and usage of the term “determinants” are vague and potentially incorrect. In the broader public health literature, references to “social determinants” convey a sense of “fundamental causes” or attributes with strong probabilistic potential to affect an outcome – usually because of stratification systems and inherent privilege or lack thereof. It seems the authors have this concept in mind, but the meaning is lost in the actual discussion of identified “determinants.” How does “aging,” “migration,” or “climate change” “cause” opportunities for decent work? Consider the vexing link between “migration” as one “determinant of decent work” and “employment creation” as one “pillar of decent work.” The absence of employment opportunities may facilitate out-migration from a region, whereas employment opportunities may facilitate in-migration. This is a fundamental principle of migration but essential to the current framework – migration is a consequence of employment opportunities, not a cause (or determinant) of employment opportunities. Similarly, age and associated human capital needed in the labor market may shape age distributions in different industries and occupations, but “aging” is not a “cause” (determinant of) decent work. The absence of clear conceptualization impedes clarity in how to use the offered framework.

 Response

Yes, the pillars also have determinant characteristics, but we are using them with the ILO meaning that they are objectives, but state that they also drive decent work. They are in contrast with the determinants (in the framework) which are broadly ‘environmental’. Language to this effect has been added.

23c. With the notable exception of “OSH Policies,” real determinants of decent work (assuming the authors mean fundamental causes) are notably absent from the framework. Again the authors are confronted with a conceptual ambiguity. That is, aren’t the pillars of decent work actually the causes of (determinants of) decent work? Strong national/regional sentiments about the importance of social protections (i.e., social welfare) theoretically create expectations that people should not have to risk injury or death to earn a living. Valuing human agency and autonomy – extended to individuals in the workplace in the form of workers’ rights creates expectations that workers should be able to oppose hazards or risks without loss of compensation or dignity. These examples highlight, again, how insufficient clarity in conceptual meaning impedes the framework’s actual utility.

Response

Language to clarify this issue has been added.

  1. There are notable WEIRD (Western, Educated, Industrialized, Rich, and Democratic) biases inherent in the framework. WEIRD countries living on “this side” of the industrial transition can easily talk about the value of OSH – in the form of worker rights and protections through social advocacy and dialogue. But how can this framework be meaningfully applied outside of WEIRD contexts? There is also a clear theme of “us versus them” – residuals of class warfare between “management” and “workers” -- in the manuscript (see especially lines 287-294). This thread, combined with other features of the manuscript (its WEIRD bias, assumptions that all risk or threats can be eliminated from jobs while creating only jobs filled with opportunities for workers to self-actualize), suggests the framework is not moving beyond classic OSH. Instead, it is rehashing and rebranding long-held beliefs and values in the field. The manuscript and the framework would be more valuable if they deliberately took on longstanding impediments to success in OSH. For example, is it reasonable to eliminate all risks from every job? Should a position be an individual’s sense of identity/actualization? Should economic growth supersede individual worker well-being? Questions like these require open consideration rather than assuming the standard OSH position of “yes,” “yes,” and “no” to each, respectively.

Response

Achieving decent work is a global goal but clearly it is not achievable everywhere at the same time. The issue that reviewer raises merits discussion but is beyond the scope of this paper. However, language has been added to raise the issue. The ‘us versus them’ interpretation reflects the real world dynamics of the work place and ‘labors’ struggle.

  1. There is relatively little “new” thinking in the manuscript. The authors cite emerging phenomena (e.g., nanoparticles), but handling them is a classic response of understanding and containing exposure. Other issues like growth in informal employment arrangements, migration, aging, and women in the labor force are hardly new – they’ve been in the literature for decades.

Response

The example of nanoparticles is in a section on new hazards hence describing the hazard was the focus. The other determinants (e.g., informal work) indeed are not new nor described as such. They are however, continuing, growing, and will influence the future of work. The innovation of the framework is that it brings all these factors together for consideration.

  1. I am sympathetic to the authors’ goal: a new framework for thinking about OSH is sorely needed. However, I remain unconvinced that the offered framework is usable because it is conceptually ambiguous and presents a biased view that perpetuates tensions among stakeholders rather than resolving them. I appreciate the broad and sweeping attempt to summarize the literature, but the broadness results in overly general statements of what is known and not known. The combination of poor conceptual clarity, inherent biases that are out of touch with developing economies (perhaps even regions [rural communities] or sectors [construction or agriculture] within developed economies), and overly generalized statements in the literature prevent this manuscript from achieving its intended goal.

Response

Yes, developing such a framework is difficult. We have tried to improve the conceptual clarity. The framework is meant to highlight and stage for the OSH community interactions between determinants and pillars where professional thinking and practical applications are needed.

Reviewer 4 Report

This manuscript presents a framework that integrates the ILO main objectives (pillars of decent work) with determinants of decent work selected by the authors. The framework is intended to organize thinking across the board array of OSH as a field. Another focus of the model is the ability to use it as work and the needs of workers evolve. This paper is very thorough and covers an incredible array of topics, ideas, and concepts. Is it well done and is extremely important to the field from both research and practice perspectives. 

There were many excellent points made in the paper. Of course, as will any framework attempting to organize an entire field, details were sparse in some cases (as appropriate). 

Would it be possible to update the employment/gender gap data given in section 3.3.1? The data are from 2018 so if they could be updated or a statement added as to why this particular year was used, that would be helpful to the reader. 

The possible upsides of wearables mentioned starting on line 398 are interesting. However, the connection to the inherent psychosocial risks of greater monitoring isn't clear from the sentences at the end of the paragraph. I suggest making the connection explicit. 

There are typos and/or missing citations in the following lines: 198, 202, 246, 574, 576, 610, 612. 

Author Response

Reviewer 4

27. This manuscript presents a framework that integrates the ILO main objectives (pillars of decent work) with determinants of decent work selected by the authors. The framework is intended to organize thinking across the board array of OSH as a field. Another focus of the model is the ability to use it as work and the needs of workers evolve. This paper is very thorough and covers an incredible array of topics, ideas, and concepts. Is it well done and is extremely important to the field from both research and practice perspectives. 

Response

Agreed

28. There were many excellent points made in the paper. Of course, as will any framework attempting to organize an entire field, details were sparse in some cases (as appropriate). 

Response

Agreed.

29. Would it be possible to update the employment/gender gap data given in section 3.3.1? The data are from 2018 so if they could be updated or a statement added as to why this particular year was used, that would be helpful to the reader. 

Response

Connection added. 

30. The possible upsides of wearables mentioned starting on line 398 are interesting. However, the connection to the inherent psychosocial risks of greater monitoring isn't clear from the sentences at the end of the paragraph. I suggest making the connection explicit. 

Response

The connection was made explicit. 

31. There are typos and/or missing citations in the following lines: 198, 202, 246, 574, 576, 610, 612. 

Response

Added citation but most seem correct.

198 – reference added
202 – typo corrected
246 – not clear what this refers to
574 – not clear what this refers to
576 – reference assessed and found to be appropriate
610 – sentence fixed
612 – reference added 

Round 2

Reviewer 2 Report

Dear Authors

I believe the quality and definitely the impact of the manuscript could be improved by naming the 'framework' and reflecting that name in the title.

Also, I am still widely in doubt whether other frameworks does not cover for the same elements, since the authors have still not sufficiently engaged with existing literature in the introductory section. However, it does not seem that the authors are decidedly wrong in anything they write.

Author Response

Comments & Responses for Paper by Schulte et al.

Reviewer 2

"Dear Authors

I believe the quality and definitely the impact of the manuscript could be improved by naming the 'framework' and reflecting that name in the title.

Response
The framework had been named.  It is the title of Figure 1: “Occupational safety and health staging framework for decent work”.  Consequently, we have modified the sentence (at the line 138) as follows:  This framework, named in the title of Figure 1, was…”.
We have also modified the title of the paper, that is currently the same of the figure 1.

Also, I am still widely in doubt whether other frameworks does not cover for the same elements, since the authors have still not sufficiently engaged with existing literature in the introductory section. 

Response
There is no other framework in the literature, as far as we know that covers such a broad range of determinants and their interactions with the pillars of decent work.  For this reason, we have consequently updated the sentence in line 140: 
There is no other framework in the literature, as far as we know, that covers such a broad range of determinants and their interactions with the pillars of decent work.

However, it does not seem that the authors are decidedly wrong in anything they write.

Response
Thank you.